

# Synergistic application of *Bacillus subtilis* IAGS174 and thiamine to mitigate salinity and lead stress in *Helianthus annuus*

Sidra Fatima[1], Waheed Ullah Khan[1], Rehana Sardar[2], Bareera Munir[1], Areeba Rehman[1], Waheed Akram[3], Iqra Munir[4] and Nasim Ahmad Yasin[3]

[1] College of Earth and Environmental Sciences, University of the Punjab, Lahore, Pakistan
[2] Department of Biological and Environmental Sciences, Emerson University, Multan, Pakistan
[3] Department of Horticulture, University of the Punjab, Lahore, Pakistan
[4] University Institute of Medical Laboratory Technology, Faculty of Allied Health Sciences, The University of Lahore, Lahore, Pakistan

Corresponding authors
Rehana Sardar,
rehana.sardar@eum.edu.pk
Nasim Ahmad Yasin,
nasimhort@gmail.com

## ABSTRACT

The increasing soil contamination with salinity and heavy metals poses serious threats to the cultivation of economically and ornamentally important plants such as *Helianthus annuus* (Sunflower). *Bacillus subtilis* IAGS174 and thiamine are well known for their role in increasing plant stress tolerance levels by multiple mechanisms. The present research aimed to assess the effect of *B. subtilis* IAGS174 and thiamine on *H. annuus* under salinity and lead (Pb) stress by analyzing the growth parameters, physiological markers, and biochemical assays. In a pot experiment, *B. subtilis* IAGS174 and thiamine were supplied to *H. annuus* plants grown in soil subjected to 500 mg/kg of salt and 150 mg/kg of Pb. The growth attributes and photosynthetic machinery of *H. annuus* plants were significantly reduced under single and combined stress of Pb and salinity. The combined stress of Pb and salinity declined the root length, shoot length, root fresh weight, shoot fresh weight, chlorophyll *a*, and chlorophyll *b* of *H. annuus* by 49%, 61%, 48%, 39%, 53%, and 55%, respectively, as compared to the control. Moreover, under stress, *H. annuus* plants exhibited higher levels of antioxidative enzymes, phenol, flavonoid and proline content. Nevertheless, the combined effect of *B. subtilis* IAGS174 and thiamine improved the fresh weight of shoots and roots, chlorophyll *a*, chlorophyll *b* and carotenoids by 34%, 38%, 15%, 18% and 16%, respectively, under the combined stress of salt and Pb to their respective controlled conditions. Supplementation of *B. subtilis* IAGS174 and thiamine significantly increased the antioxidative enzymes (superoxide dismutase, catalase, and peroxidase) and non-enzymatic antioxidants (phenol, flavonoids and proline) in sunflowers under combined and individual stress of sodium and Pb. Nevertheless, inoculation of *B. subtilis* IAGS174 accelerated the translocation of Pb and Na, while thiamine application reduced the uptake of these metals. Conclusively, single and combined application of plant growth-promoting rhizobacteria (PGPR) and thiamine proved a sustainable and effective option to improve plant tolerance against salt and Pb stress and offer new avenues for suitable agricultural practices in heavy metal and salt-contaminated soil.

## INTRODUCTION

One of the major problems agricultural land faces is the continuous increase in salinity. Around the world, salinity affects 20% of irrigated land and 7% of all arable land (*Liu et al., 2024*). India, China, the United States, Pakistan, Sudan, and Turkey have major salinized lands, and it is spreading in other countries (*Singh, 2022*). About 6.3 million hectares of land have been affected by salinity in Pakistan and approximately 4,000 hectares of fertile soil have been degraded with salinity every year (*Alobaid et al., 2025*).

The major causes of soil salinization are the weathering of rocks and climate change-induced evaporation, extensive groundwater extraction, and subsequent flooded irrigation leading to the accumulation of soluble salts in the land (*Maryam et al., 2023*). Anthropogenic activities that cause an increase of lead (Pb) content in soil and water are industrial processes, transportation, fertilizer, pesticide use and irrigation of soil with industrial effluent (*Ijaz et al., 2020*). Nations, including Pakistan, have stated that Pb concentrations in the soil have risen drastically over the last few decades and decreased crop yields (*Altaf et al., 2021*). This high salinity and Pb level have an unfavorable influence on germination, plant development, and production of reactive oxygen species (ROS), which results in plant death by negatively affecting several metabolic processes like photosynthesis, respiration, transpiration, membrane characteristics, cellular hemostasis, and hormone balance, (*Patwa et al., 2024*; *Naz et al., 2025*). High salt concentration also has an osmotic and ion-specific effect on plant growth, oxidative stress tolerance and yield. Since most metals have negative effects at low doses (between one and 10 mg/mL), thus heavy metal and salinity pollution have become a serious concern for plant growth (*Rojas-Solis, Larsen & Lindig-Cisneros, 2023*).

For the soil remediation from these pollutants, many physical, chemical, and biological techniques are being used all across the world. Physicochemical methods are mostly costly, unfriendly, and impractical when handling larger amounts of soil. On the other hand, bioremediation techniques such as using growth-promoting rhizobacteria (PGPRs) in combination with plants are considered a feasible, economic and eco-friendly approach for the remediation of multi-contaminated soil (*Sarwar et al., 2023*). The PGPRs are found near plant roots and improve plant growth by providing nutrients and growth-promoting hormones (*Shabaan et al., 2023*). Hyper-accumulator plants can uptake 100 times more heavy metals from soil and are deemed a viable option to decontaminate salt and Pb-polluted sites (*Altaf et al., 2021*). Sunflower (*Helianthus annuus*) is a hyperaccumulator and the fourth-largest global commercial oilseed crop. Over the past 20 years, its production has increased significantly, but since 2010, its production has decreased steadily due to different causes, including soil degradation with salinity and heavy metals (*Aftab et al., 2020*). According to many investigations, most PGPR strains like *Agrobacterium*, *Bacillus*, and *Rhizobium* species increase the host plants' ability to withstand heavy metal toxicity. Numerous procedures such as fixation of atmospheric nitrogen, the phosphate solubility and the synthesis of phytohormones (such as gibberellin, abscisic acid, and indole 3-acetic acid (IAA), 1-aminocyclopropane-1-carboxylate (ACC) deaminase, siderophore synthesis, and exopolysaccharide (EPS), can be used by these PGPRs to improve plant growth and

heavy metal stress tolerance in plants (*Anwar et al., 2024*). Furthermore, PGPRs increase the enzymatic and non-enzymatic antioxidative activity to confer salinity and Pb toxicity in plants (*Hahm et al., 2017*).

According to recent research, apart from plant growth regulators (PGRs), vitamins like thiamine and ascorbic acid also help the plants to deal with the deadly consequences of both biotic and abiotic stressors by participating in energy-producing mechanisms like the Krebs' cycle and the Calvin cycle. The foliar application of thiamine was found to dramatically boost growth, photosynthetic pigments, and antioxidant activities in many plants grown under abiotic stress regimes (*Naheed et al., 2021*). Thiamine contributes to the defensive network by acting as an antioxidant or indirectly by supplementing NADH and NADPH in plants. Thiamine supplementation may boost the performance of an anti-oxidative defense system in plants to cope with salinity and Pb toxicity (*Sanjari et al., 2019*).

Researchers have reported the beneficial role of individual application of PGPRs and thiamine on numerous plant species under a single stress condition. However, there is a lack of comprehensive research on the co-application of PGPR and thiamine to a hyperaccumulator plant like *H. annuus* (sunflower) subjected to combined salinity and Pb stress. The main objectives of this study were: **(i)** evaluation of the potential of *B. subtilis* IAGS174 and thiamine for improvement of growth, photosynthetic pigments, enzymatic and non-enzymatic antioxidants, and attenuation of lipid peroxidation in sunflower exposed to Pb and salinity stress and **(ii)** the assessment of *B. subtilis* IAGS174 and thiamine role in the translocation of sodium (Na) and Pb from soil to sunflower tissues. This study unveils a new approach for the synergistic role of *B. subtilis* IAGS174 and thiamine for sustainable crop production and stress tolerance in plants subjected to multiple stresses.

## MATERIAL AND METHODS

The pot experiment was conducted in the wirehouse at the College of Earth and Environmental Sciences, University of Punjab, Lahore (31.5204°N, 74.3587°E).

### Inoculum preparation and soil inoculation

To prepare inoculation, a loopful of rhizobacteria (*B. subtilis* IAGS174) was inoculated into Luria Bertani (LB) broth media, which was then incubated in a mechanical shaker for 2 days at 27 °C. The optical densities were then measured to establish a uniform population of $10^8$–$10^9$ CFU mL$^{-1}$. Approximately 8–10 mL of bacterial culture was applied to the soil with the help of a sterile syringe

### Pot experiment

A pot experiment was performed to determine the positive effect of rhizobacteria and thiamine on Pb and salt stress in sunflower plants. The plastic pots were filled with soil (1.5 kg) obtained from the agricultural fields of the University of the Punjab, Pakistan. Salt and heavy metal stress were applied by amending sodium chloride (NaCl) (500 mg/kg soil) and Pb $(NO_3)_2$ (150 mg/kg) in the soil. The pots were moistened up to field capacity

and then kept for 14 days to blend up the pollutants in the soil. The sunflower seeds were procured from the Roshan Seed shop in Lahore, Pakistan. For surface sterilization, seeds were immersed in 70% ethanol for 30 s and 1% sodium hypochlorite for an additional 60 s, following multiple washes with distilled water. The healthy seeds (eight to ten) were planted in each pot, and thinning was performed after germination to keep five plants in each pot. After three weeks of planting, growth regulator (PGPR, thiamine) treatments were applied. The bacterial inoculum (10 mL) was drenched into the soil and thiamine (100 ppm) was supplemented by foliar application to the sunflower plants. The experiment was done in a wirehouse. The environmental conditions in the wirehouse were a temperature of 28–35 °C, a night/day period of 11/13 h, and humidity of $75 \pm 5\%$. The plants were watered at regular intervals to keep optimum moisture in the soil. Each treatment was replicated thrice by a completely randomized design (CRD). A total of 48 pots for 16 treatments were planted with sunflower seeds, and these plants were uprooted after 120 days. Hence, total of 16 treatments was planned by using either single or combined treatments of both growth promoters (microbe, thiamine) and single and co-application of pollutants (Na, Pb) such as C; control, B; Bacteria, T; thiamine, B+T; Bacteria+ thiamine, S; Salinity, S+B; Salinity+ Bacteria, S+T; Salinity and Thiamine, S+B+T; Salinity and Bacteria +Thiamine, Pb; lead, Pb+B; Lead+Bacteria, Pb+T; Lead+Thiamine, Pb+B+T; Lead+Bacteria+Thiamine, Pb+S; Lead+ Salinity, Pb+S+B; Lead+Salinity+Bacteria, Pb+S+T; Lead+Salinity+Thiamine, and Pb+S+B+T; Lead+Salinity+Bacteria+Thiamine.

## Assessment of photosynthetic pigments

The fully mature and widened fresh leaves from the center of each plant were selected and removed. To estimate chlorophyll (Chl.) pigments, 0.25 g of fresh leaf tissue was blended with 80% acetone to make a plant extract. After extraction, the mixtures were centrifuged at $12,000\times$ g for 10 min. The amount of Chl. *a* and Chl. *b* was measured by monitoring the absorbance of the supernatant at wavelengths of 663 nm and 645 nm, respectively, with the help of a spectrophotometer (UV/VIS, Cecil Aquarius CE 7200). The relative absorbance of the supernatant was obtained at 480 nm for examining carotenoid content (*Liu et al., 2024*). The formulas used for calculating the chlorophyll content are:

$$\text{Chl. 'a' (mg g}^{-1}\text{ FW)} = \frac{(12.7 \times A663) - (2.69 \times A645) \times \text{Sample volume}}{1000 \times W}$$

$$\text{Chl. 'b' (mg g}^{-1}\text{ FW)} = \frac{(12.7 \times A645) - (4.68 \times A663) \times \text{Sample volume}}{1000 \times W}$$

$$\text{Total chlorophyll (mg g}^{-1}\text{ FW)} = \frac{20(A645) + 8.02(A663) \times \text{Sample volume}}{1000 \times W}$$

$$\text{Carotenoids (mg g}^{-1}\text{ FW)} = OD480 + 0.114\,(OD663) - 0.638(OD645).$$

## Antioxidant enzymes analysis

Enzyme extract was obtained by adding one g of the plant's fresh leaves to a mortar and crushing with a pestle while pouring liquid $N_2$. Afterwards, the plant biomass was homogenized in 2.0 mL of an ice-cold 50 mM sodium phosphate buffer. The homogenous mixture was centrifuged at $10,000\times$ g for 20 min at 4 °C. The resulting supernatant was

employed to assess different antioxidative enzymes with the help of a spectrophotometer (*Rao & Sresty, 2000*). Each analysis was replicated thrice.

Superoxide dismutase (SOD, EC: 1.5.1.1 ) content was observed by pouring 0.1 mL of enzyme extract into 1.5 mL 50 mM sodium phosphate (pH 7.8), 0.300 mL 750 μM nitro blue tetrazolium (NBT), 0.300 mL 20 μM riboflavin, 0.3 mL 130 μM methionine, 0.3 mL 100 μM EDTA-N Spectrophotometer (UV/VIS, Cecil Aquarius CE 7200) was used to note reading at 560 nm after the reaction mixture was illuminated under light of 4,000 flux for 20 min (*Bin et al., 2010*). Peroxidase (POD, EC: 1.11.1.7) activity was analyzed using 50 μL enzyme extract and 1.0 mL 0.3% $H_2O_2$, 1.0 mL 50 mM sodium phosphate (pH 5.5), and 0.95 mL 0.2% guaiacol. This was recorded at 470 nm using a spectrophotometer (UV/VIS, Cecil Aquarius CE 7200). A reaction mixture containing 500 μL 0.1 M $H_2O_2$, 3.0 mL 50 mM sodium phosphate (pH 7.8), 1.0 mL deionized water, and 200 μL enzyme extract was prepared for catalase (CAT, EC: 1.11.1.6) activity. A spectrophotometer (UV/VIS, Cecil Aquarius CE 7200) was used to note the absorbance at 240 nm (*Rhaman et al., 2024*).

## Quantification of hydrogen peroxide and malondialdehyde

For malondialdehyde (MDA) determination, 0.2 g of plant leaf sample was crushed in five mL of 0.1% TCA and centrifuged for 5 min at 10,000 g. The reaction mixture was made by adding one mL of the supernatant aliquot to four mL of 20% TCA with 0.5% thiobarbituric acid (TBA). The absorbance of the reaction mixture was estimated at 532 nm by a spectrophotometer (UV/VIS, Cecil Aquarius CE 7200), and the reading for non-specific absorption at 450 nm and 600 nm was subtracted (*Zhang et al., 2013*. The MDA level (mol $g^{-1}$ FW) was recorded using (= 155 $mM^{-1}$ $cm^{-1}$). The hydrogen peroxide ($H_2O_2$) content was estimated by crushing 0.5 g of fresh plant leaf with five mL of trichloroacetic acid (TCA 0.1%) and then centrifuging at 12,000 g for 20 min at 4 °C. Afterward, 0.5 mL of supernatant liquor was homogenized with 0.5 mL of 10 mM $KPO_4$ buffer. The reading of absorbance was noted at 390 nm by a spectrophotometer (UV/VIS, Cecil Aquarius CE 7200) (*Velikova, Yordanov & Edreva, 2000*).

## Estimation of flavonoid and phenolic content

A volumetric flask was filled with one mL of plant extracts and four mL of distilled water to determine the flavonoid concentration. The solution was mixed with 0.3 mL of 5% $NaNO_2$ after 5 min. 0.3 mL of 10% $AlCl_3$ was added to it. After 6 min, two mL of 1 M NaOH was added to the solution. Distilled water was added to make the volume 10 mL. The absorbance of the solution was measured at 510 by a spectrophotometer (UV/VIS, Cecil Aquarius CE 7200) (*Zhishen, Mengcheng & Jianming, 1999*). Phenol contents were determined by immersing two g of plant leaf in 10 mL of 80% methanol for 15 min at 65 °C. Afterwards, one mL of plant extract was mixed with 250 μL of Folin-Ciocalteau reagent (1N) and five mL of distilled water. The reaction mixture was stored at 30 degrees Celsius. To identify the exact number of phenols, the absorbance at 725 nm was measured by a spectrophotometer (UV/VIS, Cecil Aquarius CE 7200) and compared to the gallic acid curve (*Zieslin & Ben-Zaken, 1993*).

## Proline estimation

A sample of dried and prewashed leaf (100 mg) was placed into a flask containing 10 mL of sulfosalicylic acid (3%). The mixture was vortexed and then filtered through Whatman's no-filter paper. Then, two mL of the filtrate was transferred to a glass tube containing as much glacial acetic acid as there was acid ninhydrin. It was placed in a water bath at 100 °C for 0.5 h. Then, four mL of toluene was added, and the chromophore was aspirated. This reaction mixture was incubated at 25 °C for 0.5 h, and its colorimetric value was recorded at 520 nm by a spectrophotometer (UV/VIS, Cecil Aquarius CE 7200) and compared to the standard curve (*Bates, Waldren & Teare, 1973*).

## Plant growth and biomass production assessment

After 120 days, the plants were removed from the pots and properly rinsed with distilled water. Plant parts, including roots, shoots, and leaves, were separated. The lengths of the roots and shoots were measured. The values of fresh biomass production from root and shoot were also measured using an electrical balance. The root and shoot samples were placed in an oven at 700 °C for one day. The root and shoot dry weights were also calculated.

## Heavy metal analysis

The dried plant samples were ground into a fine powder, and 1 g was mixed with 10 mL of $HNO_3$ and five mL of $HCLO_4$. The mixture was put in a flask and transferred to the hot plate, which was then placed in the hot plate at 1,500 °C until the solution turned transparent and reduced to five mL. After the digestion, 45 mL of distilled water was mixed into this solution, making a solution of 50 mL. The solution was filtered twice using the Whatman 42 filter paper. The filtrate was shifted to sample bottles, and distilled water was poured to make the sample up to 100 mL. The Pb quantity was estimated using an atomic absorption spectrophotometer (AAS; PerkinElmer AAnalyst 800; PerkinElmer, Waltham, MA, USA), while Na was analyzed by a flame photometer (Model PFP7, 90–125 V).

The metal tolerance index, translocation factor (TF) and bio-concentration factor (BCF) were calculated by the methodology proposed by *Wu et al. (2018)*. The metal tolerance index was analyzed for the evaluation of the ability of the plant to grow in the presence of metal concentration by a formula:

$$\text{Metal tolerance index} = \frac{\text{Mass of treated plant}}{\text{Mass of control plant}} \times 100$$

The translocation factor was determined as;

$$\text{TF} = \frac{\text{or Na conc. in shoot}}{\text{Pb or Na conc. in root}}$$

The bio-concentration factor (BCF) was determined by:

$$\text{BC} = \frac{\text{Pb or Na conc. in shoot}}{\text{or Na conc. in soil}}.$$

## Statistical analysis

The experiment was performed with three replicates of each treatment in a completely randomized design (CRD). The obtained results were demonstrated as mean ±standard deviation. For statistical analysis, a one-way analysis of variance (ANOVA) was performed with the help of DSAASTAT software. Significance among multiple treatments was checked by performing Duncan's multiple range test (DMRT) at $P \leq 0.05$ significance level using DSAASTAT software.

## RESULTS

### Effect *B. subtilis* IAGS174 and thiamine on growth parameters of *H. annuus* subjected to salinity and Pb stress

The effects of different treatments were significant on shoot length (SL), root length (RL), shoot fresh weight (SFW), shoot dry weight (SDW), root fresh weight (RFW), and root dry weight (RDW) of sunflower. Both salinity and Pb stress and their combination significantly hampered shoot and root length, dry weight, and fresh weight of the shoot and root of sunflower compared to the control treatment. The Pb, S and Pb+S treatments declined the shoot length of sunflower by 38%, 25% and 49% respectively as compared to untreated control. However, the individual application of PGPR and thiamine and their combined treatment enhanced the growth of sunflower plants under normal and stress conditions. The efficacy of *B. subtilis* IAGS174 and thiamine on sunflower was proved better under un-stressed conditions. The combined B+T treatment under Pb and salt stress was found to be very effective and meaningfully improved shoot length, root length, shoot fresh weight, shoot dry weight, root fresh weight, and root dry weight by 28.12%, 60%, 6.84%, 9.3%, 16.3% and 32.25% as compared to Pb+S treatment (Table 1).

### Influence *B. subtilis* IAGS174 and thiamine on photosynthetic attributes of *H. annuus* subjected to salinity and Pb stress

Both Pb and salinity stress either single or combined significantly reduced the Chl. *a*, Chl. *b*, total chlorophyll and carotenoid content in sunflower plants. Combined Pb+S stress significantly (DMRT at $p \leq 0.05$) reduced Chl. *a* by 44.5%, Chl. *b* by 54.81%, total Chl. by 47.33%, and carotenoid by 45.21% as compared to the untreated control. However, the combined application of bacteria and thiamine (B+T) treatment on sunflower under Pb and salt stress was the most effective in improving the Chl. *a*, Chl. *b*, total chlorophyll and carotenoid content by 16.33%, 16.39%, 15.66%, and 14.28%, respectively, as compared to the Pb+S treatment (Table 2).

### Impact of *B. subtilis* IAGS174 and thiamine on antioxidant enzymes of *H. annuus* subjected to salinity and Pb stress

The antioxidant enzyme parameters like SOD, POD, and CAT showed a significant change in sunflower under all the treatments. The amount of SOD, POD, and CAT showed a significant (DMRT at $p \leq 0.05$) increase in sunflower under salt (220.5%, 226.58%, and 220%, respectively) and Pb stress (412.8%, 358.38%, and 97.14%) as compared to the control. While combined Pb+ S treatment significantly (DMRT at $p \leq 0.05$) increased
**Table 1 Impact of *Bacillus subtilis* IAGS174 and thiamine on growth attributes of *H. annuus* subjected to Pb and salinity stress.**

| Treatments | Growth attributes | | | | | |
|---|---|---|---|---|---|---|
| | Shoot L (cm) | Root L (cm) | SFW (g/plant) | SDW g/plant) | RFW (g/plant) | RDW (g/plant) |
| C | 62 ± 1.63bcde | 9.1 ± 1.63c | 10.2 ± 1.39cd | 3.15 ± 0.03d | 1.78 ± 0.06b | 0.61 ± 0.02d |
| B | 68 ± 0.85ab | 12.2 ± 1.59ab | 14.5 ± 1.23a | 3.53 ± 0.02b | 1.92 ± 0.06a | 0.71 ± 0.02b |
| T | 65 ± 1.63bc | 11.5 ± 0.66b | 13.1 ± 1.15b | 3.43 ± 0.02c | 1.86 ± 0.03ab | 0.65 ± 0.05c |
| B+T | 72 ± 2.45a | 13.4 ± 0.54a | 15.3 ± 1.01a | 4.36 ± 0.03a | 1.97 ± 0.02a | 0.78 ± 0.05a |
| S | 47 ± 1.63h | 5.4 ± 0.24fg | 8.5 ± 0.86efj | 2.75 ± 0.02f | 1.34 ± 0.04c | 0.51 ± 0.10h |
| S+B | 60 ± 2.45cdef | 8.6 ± 0.16cd | 9.1 ± 0.24cdef | 2.88 ± 0.02e | 1.47 ± 0.03c | 0.57 ± 0.04f |
| S+T | 56 ± 2.16ef | 7.2 ± 0.29de | 8.9 ± 0.08defg | 2.85 ± 0.06e | 1.41 ± 0.04c | 0.54 ± 0.02g |
| S+B+T | 63 ± 2.45bcd | 9.4 ± 0.29c | 10.5 ± 0.16c | 2.9 ± 0.05e | 1.51 ± 0.03c | 0.59 ± 0.07e |
| Pb | 39 ± 2.45ig | 4.1 ± 0.37gh | 8.1 ± 0.22efg | 2.31 ± 0.01g | 1.13 ± 0.02c | 0.43 ± 0.02k |
| Pb+B | 54 ± 2.16fg | 8.3 ± 0.43cd | 9.2 ± 0.17fcde | 2.45 ± 0.02g | 1.25 ± 0.02c | 0.46 ± 0.01j |
| Pb+T | 49 ± 2.94gh | 6.8 ± 0.37ef | 8.7 ± 0.16defg | 2.39 ± 0.02g | 1.21 ± 0.05c | 0.45 ± 0.02j |
| Pb+B+T | 58 ± 2.44def | 8.9 ± 0.67c | 9.3 ± 0.16fcde | 2.53 ± 0.01g | 1.38 ± 0.02c | 0.49 ± 0.02i |
| Pb+S | 32 ± 5.10j | 3.5 ± 0.33h | 7.3 ± 0.08g | 2.15 ± 0.02g | 0.92 ± 0.02c | 0.31 ± 0.02k |
| Pb+S+B | 39 ± 3.74ig | 5.3 ± 0.33fg | 7.6 ± 0.35fg | 2.28 ± 0.09g | 0.98 ± 0.08c | 0.38 ± 0.05k |
| Pb+S+T | 37 ± 5.09ig | 4.7 ± 0.24gh | 7.3 ± 0.16g | 2.32 ± 0.02g | 0.96 ± 0.04c | 0.36 ± 0.03k |
| Pb+S+B+T | 41 ± 3.74i | 5.6 ± 0.24fg | 7.8 ± 0.08efg | 2.35 ± 0.01g | 1.07 ± 0.18c | 0.41 ± 0.02k |

**Notes.**
Values are means ± SD of three replicates. Different letters in a column depict significant differences among treatments according to DMRT at $p \leq 0.05$. C, control; B, bacteria; T, thiamine; Pb, Lead stress and S, Salt stress.

the antioxidant machinery of the sunflower plant by 6.62-fold in SOD, 5-fold on POD, and 3.2-fold in CAT as compared to the control. The application of PGPR to sunflower exposed to Pb and salinity stressed soil further increased the antioxidative enzymes., The application of thiamine also augmented the antioxidant enzyme levels in sunflower plants under stressed regimes compared to the relevant control, but less than the bacterial and B+T treatments. The B+T treatment significantly (DMRT at $p \leq 0.05$) improved the activity of SOD, POD, and CAT in sunflower plants growing in Pb and salt-stressed soil by 67.74%, 65.08%, and 34.82%, respectively, as compared to Pb+S treatment (Fig. 1).

### Effect of *B. subtilis* IAGS174 and thiamine on phenol, flavonoid and proline contents of *H. annuus* exposed to salinity and Pb stress

The sunflower plants exhibited an increasing trend in phenol, flavonoid, and proline contents under single or combined Pb and salinity stress regimes. However, PGPR and thiamine application alone or in combination further enhanced phenol, flavonoid, and proline content of plants exposed to Pb and Na stress as compared to the control plants. The combined Pb+S treatment significantly (DMRT at $p \leq 0.05$) increased phenol content by 84.13%, flavonoid content by 2.14-fold, and proline content by 3.31-fold. Combined application of B+T under stress (Pb+S) conditions also resulted in a substantial (DMRT at $p \leq 0.05$) increase in phenol, flavonoid, and proline content by 38%, 45% and 49% as compared to the respective control (Table 2).

**Table 2** Impact of *Bacillus subtilis* IAGS174 and thiamine on photosynthetic attributes (chlorophyll *a*, *b*, total chlorophyll, and carotenoids) and biochemical parameter (phenol, flavonoid, and proline) of *H. annuus* subjected to Pb and salinity stress.

| Treatments | Chl. *a* | Chl. *b* | Total Chl. | Carotenoids | Phenol | Flavonoids | Proline |
|---|---|---|---|---|---|---|---|
| | mg g$^{-1}$ FW | mg g$^{-1}$ FW | mg g$^{-1}$ FW | mg g$^{-1}$ FW | mg(GAE)/g FW | mg(GAE)/g FW | mmol kg-1 DW |
| C | 2.76 ± 0.10abcd | 1.35 ± 0.45abcd | 4.12 ± 1.68bc | 1.15 ± 0.47ab | 37.2 ± 1.7j | 12.3 ± 1.81j | 12.65 ± 0.51i |
| B | 3.48 ± 0.16ab | 1.82 ± 0.25ab | 5.3 ± 2.16a | 1.64 ± 0.67ab | 44.5 ± 2.3ij | 14.1 ± 0.68j | 12.92 ± 0.64i |
| T | 3.29 ± 0.17abc | 1.73 ± 0.71abc | 5.05 ± 2.06ab | 1.53 ± 0.62ab | 42.7 ± 2.1ij | 13.4 ± 0.71j | 12.71 ± 0.63i |
| B+T | 3.75 ± 0.16a | 1.92 ± 0.62a | 5.69 ± 2.31a | 1.79 ± 0.57a | 46.1 ± 2.2i | 15.2 ± 0.61j | 13.54 ± 0.65i |
| S | 2.14 ± 0.82abcd | 0.92 ± 0.36abcd | 3.09 ± 1.33d | 0.91 ± 0.29ab | 54.8 ± 2.4h | 17.9 ± 0.87i | 27.25 ± 1.31h |
| S+B | 2.62 ± 1.07abcd | 0.98 ± 0.75abcd | 3.8 ± 1.9bc | 1.12 ± 0.44ab | 67.6 ± 3.1ef | 23.4 ± 1.2fg | 29.62 ± 1.45gh |
| S+T | 2.56 ± 1.15abcd | 0.93 ± 0.37abcd | 3.49 ± 1.13cd | 1.05 ± 0.44ab | 63.2 ± 2.9fg | 21.5 ± 1.1gh | 28.93 ± 1.44gh |
| S+B+T | 2.71 ± 1.10abcd | 1.06 ± 0.03abcd | 3.79 ± 1.26c | 1.18 ± 0.59ab | 71.5 ± 3.5de | 25.2 ± 1.3ef | 31.43 ± 1.57fg |
| Pb | 1.92 ± 0.10bcd | 0.75 ± 0.31cd | 2.68 ± 1.10de | 0.83 ± 0.41ab | 59.8 ± 2.29gh | 20.3 ± 0.75hi | 32.36 ± 1.37fg |
| Pb+B | 2.28 ± 0.09abcd | 0.84 ± 0.36bcd | 3.15 ± 1.28cd | 0.95 ± 0.38ab | 74.9 ± 3.1cd | 27.2 ± 1.72de | 36.82 ± 1.50de |
| Pb+T | 2.23 ± 0.74abcd | 0.81 ± 0.40bcd | 3.07 ± 1.27cde | 0.87 ± 0.38ab | 69.6 ± 2.7def | 23.4 ± 1.1fg | 34.97 ± 1.75ef |
| Pb+B+T | 2.36 ± 0.76abcd | 0.89 ± 0.35abcd | 3.31 ± 1.36cde | 0.98 ± 0.49ab | 78.3 ± 3.4c | 29.6 ± 1.3d | 39.81 ± 1.79cd |
| Pb+S | 1.56 ± 0.61d | 0.62 ± 0.24d | 2.17 ± 0.76e | 0.63 ± 0.31b | 68.5 ± 3.2def | 26.4 ± 1.4e | 41.95 ± 2.01c |
| Pb+S+B | 1.69 ± 0.84cd | 0.67 ± 0.26d | 2.39 ± 0.83e | 0.68 ± 0.34b | 85.2 ± 4.1b | 35.7 ± 1.9b | 63.74 ± 3.187a |
| Pb+S+T | 1.64 ± 0.75cd | 0.65 ± 0.26fd | 2.31 ± 0.98e | 0.65 ± 0.32b | 79.1 ± 3.9c | 32.6 ± 1.7c | 57.92 ± 2.84b |
| Pb+S+B+T | 1.79 ± 0.73cd | 0.73 ± 0.28cd | 2.51 ± 1.02de | 0.75 ± 0.29ab | 94.7 ± 4.57a | 38.3 ± 1.8a | 61.93 ± 2.85a |

Notes.
Values are means ± SD of three replicates. Different letters in a column depict significant difference among treatments according to DMRT at $p \leq 0.05$. C, control; B, bacteria; T, thiamine; Pb, Lead stress and S, Salt stress.

## Influence of *B. subtilis* IAGS174 and thiamine on malondialdehyde and hydrogen peroxide levels of *H. annuus* exposed to salinity and Pb stress

Single or combined Pb and salinity stress attenuated the levels of MDA and $H_2O_2$ in sunflower compared with the untreated control. Lead and salt stress significantly (DMRT at $p \leq 0.05$) increased the MDA and $H_2O_2$ levels in sunflower plants by 2.28-fold and 2.08-fold in the Pb+S treatment compared to the relevant control. The *B. subtilis* IAGS174 and thiamine-assisted *H. annuus* plants exhibited a decline in the MDA and $H_2O_2$ contents during toxic regimes. The treatment, including PGPR and thiamine in combination, significantly (DMRT at $p \leq 0.05$) reduced the MDA and $H_2O_2$ levels in sunflower plants raised under Pb and salt stress by 26.57% and 39.70%, respectively, as compared to the Pb+ S treatment (Fig. 2).

## Effect of *B. subtilis* IAGS174 and thiamine on lead and sodium uptake in *H. annuus*

Under salt and Pb stress conditions, sodium and Pb uptake was observed in root and shoot tissues of sunflower. However, the PGPR inoculation enhanced the Na uptake, while thiamine application reduced the Na uptake. The amount of salt was much higher when treated with PGPR than with thiamine, as PGPR acts as a phyto-extractant, which tends to absorb salt from the soil. Thiamine does phyto-stabilization by keeping the metal-stabilized near roots, preventing them from entering the plants (Table 3).

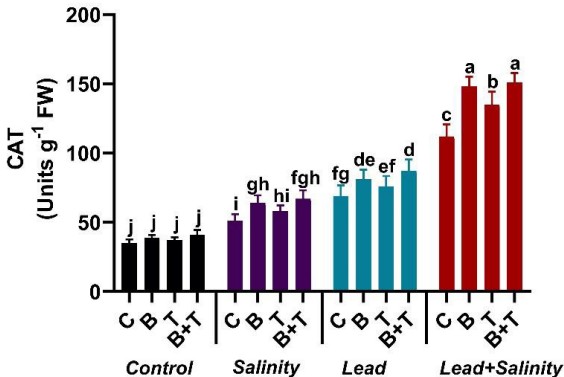

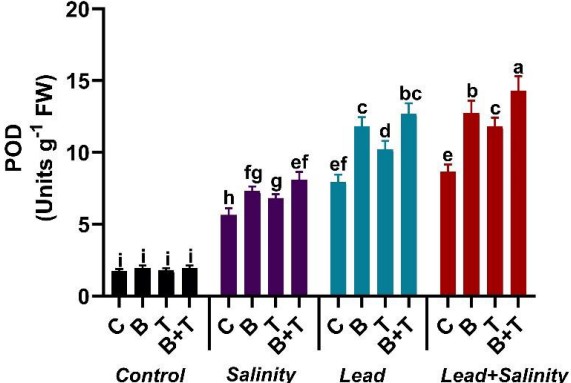

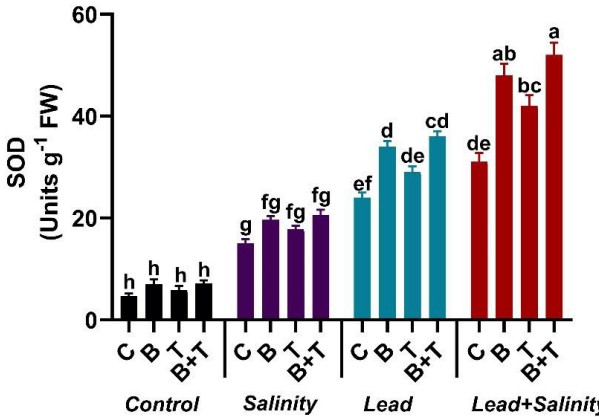

**Figure 1  Effect of *Bacillus subtilis* IAGS174 and thiamine on antioxidant enzymes of *H. annuus* subjected to Pb and salinity stress.** Values presented are means+SD (*n* = 3). The different letters in a column show significant difference among the treatments at *p* > 0.05 (DMRT). C, control; B, bacteria; T, thiamine.

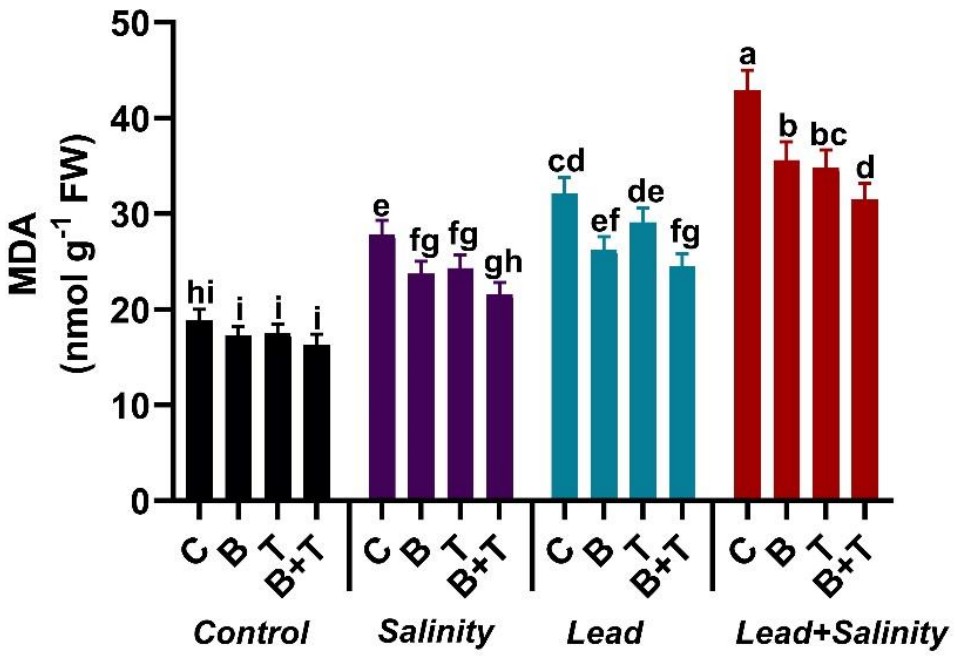

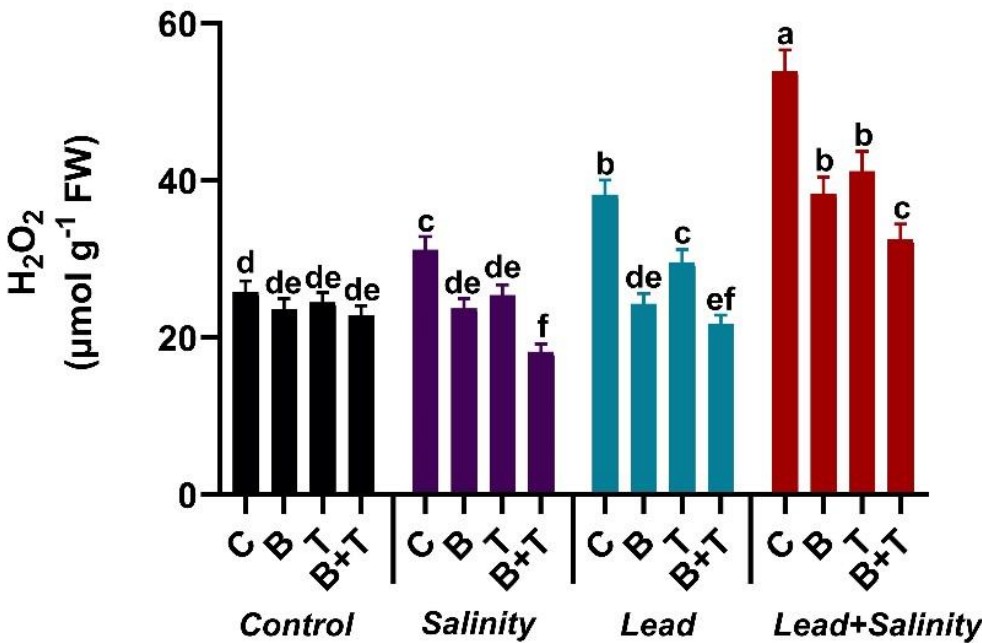

**Figure 2** **Effect of *Bacillus subtilis* IAGS174 and thiamine on malondialdehyde (MDA) and hydrogen peroxide (H₂O₂) level of *H. annuus* subjected to Pb and salinity stress.** Values presented are means+SD ($n = 3$). The different letters in a column show significant difference among the treatments at $p > 0.05$ (DMRT). C, control; B, bacteria; T, thiamine.

**Table 3** Influence of *Bacillus subtilis* IAGS174 and thiamine on Na and Pb uptake, translocation factor, bioconcentration factor and metal tolerance index of *H. annuus* plants subjected to Pb and salinity stress.

| Treatments | Sodium (Na) uptake | | | | | Lead (Pb) uptake | | | | |
|---|---|---|---|---|---|---|---|---|---|---|
| | Root (Na) mg kg$^{-1}$ | Shoot (Na) mg kg$^{-1}$ | TF | BCF | MTI | Root (Pb) mg kg$^{-1}$ | Shoot (Pb) mg kg$^{-1}$ | TF | BCF | MTI |
| C | 1.51 ± 0.16e | 1.72 ± 0.11g | 1.13cd | 0.003de | 100cd | ND | ND | – | – | – |
| B | 1.69 ± 0.22e | 1.89 ± 0.17g | 1.12d | 0.004d | 110bc | ND | ND | – | – | – |
| T | 1.48 ± 0.22e | 1.53 ± 0.25g | 1.03f | 0.003de | 89e | ND | ND | – | – | – |
| B+T | 1.62 ± 0.24e | 1.71 ± 0.23g | 1.06ef | 0.003de | 99d | ND | ND | – | – | – |
| S | 41 ± 10.23bc | 57 ± 9.90cd | 1.40ab | 0.114ab | 100cd | ND | ND | – | – | – |
| S+B | 51 ± 3.27a | 73 ± 3.74a | 1.43a | 0.146a | 128a | ND | ND | – | – | – |
| S+T | 29 ± 4.55d | 38 ± 5.35f | 1.31bc | 0.076bc | 67ef | ND | ND | – | – | – |
| S+B+T | 48 ± 3.74ab | 65 ± 5.1b | 1.35b | 0.13ab | 114b | ND | ND | – | – | – |
| Pb | 0.85 ± 0.22e | 0.93 ± 0.22g | 1.09e | 0.002e | 100cd | 91 ± 3.74c | 157 ± 2.94d | 1.73b | 1.05bc | 100bc |
| Pb+B | 1.02 ± 0.25e | 1.14 ± 0.07g | 1.12d | 0.002e | 122ab | 119 ± 4.55a | 215 ± 3.27a | 1.81a | 1.43a | 137a |
| Pb+T | 0.79 ± 0.25e | 0.82 ± 0.08g | 1.04f | 0.002e | 88e | 74 ± 2.94e | 113 ± 5.10f | 1.53c | 0.75d | 72d |
| Pb+B+T | 0.91 ± 0.10e | 0.98 ± 0.36g | 1.08ef | 0.002e | 105c | 108 ± 5.10b | 191 ± 4.55b | 1.77ab | 1.27ab | 122ab |
| Pb+S | 37 ± 4.9c | 51 ± 4.55de | 1.38ab | 0.102b | 100cd | 83 ± 4.55d | 139 ± 6.68e | 1.68bc | 0.93c | 100bc |
| Pb+S+B | 41 ± 4.32bc | 59 ± 4.55bc | 1.44a | 0.118ab | 116b | 96 ± 5.89c | 168 ± 5.89c | 1.75ab | 1.12b | 121ab |
| Pb+S+T | 27 ± 5.1d | 32 ± 2.94f | 1.19cd | 0.064c | 63f | 71 ± 3.74e | 109 ± 4.32f | 1.54c | 0.73d | 78c |
| Pb+S+B+T | 39 ± 4.55c | 48 ± 4.55e | 1.23c | 0.096bc | 94de | 92 ± 9.63c | 158 ± 4.24d | 1.71bc | 1.05bc | 114b |

**Notes.**
Values are means ± SD of three replicates. Different letters in a column depict significant difference among treatments according to DMRT at $p \leq 0.05$. C, control; B, bacteria; T, thiamine; Pb, Lead stress and S, Salt stress. TF, Translocation factor; BCF, Bio-concentration factor; MTI, Metal tolerance index.

A similar trend was observed with Pb concentration in the root and shoot of the sunflower. Under thiamine influence, Pb uptake was reduced by 28% in the shoot and 18.6% in the root of the sunflower. After PGPR inoculation, significantly (DMRT at $p \leq 0.05$) enhanced Pb uptake was observed in the root and shoot of sunflower by 30.7% and 36.94%, respectively, as compared to the Pb treatment. The combined treatment (B+T) showed high translocation values but less than the PGPR treatment. Similarly, increased translocation of Pb and Na wereobserved in *B. subtilis* IAGS174 assisted sunflower exposed to Pb+S treatment, while reduced translocation was noted applying thiamine. Table 3 shows the TF, BCF, and MTI values for thiamine and bacteria/PGPR strain (Table 3).

## Principal component analysis

The biplot (loading and score) resulted from the principal component analysis (PCA) to evaluate the efficacy of PGPR strain and thiamine-induced ameliorative effects on the biochemical and physiological attributes of sunflower (*H. annuus*). PCA of plants exposed to Pb and salt stress are given in Fig. 3. The first two components of PCA, *i.e.,* PC1 and PC2, showed maximum contribution and accounted for 86.5% of the total variance in the given database. The PC1 accounted for a 75.8% variance, while PC2 accounted for a 10.7% variance in the given dataset, correspondingly. All 16 treatments were dispersed successfully by the first two principal components. The distribution pattern provided a clear indication that bacterial strain and thiamine supplementation under combined (Pb

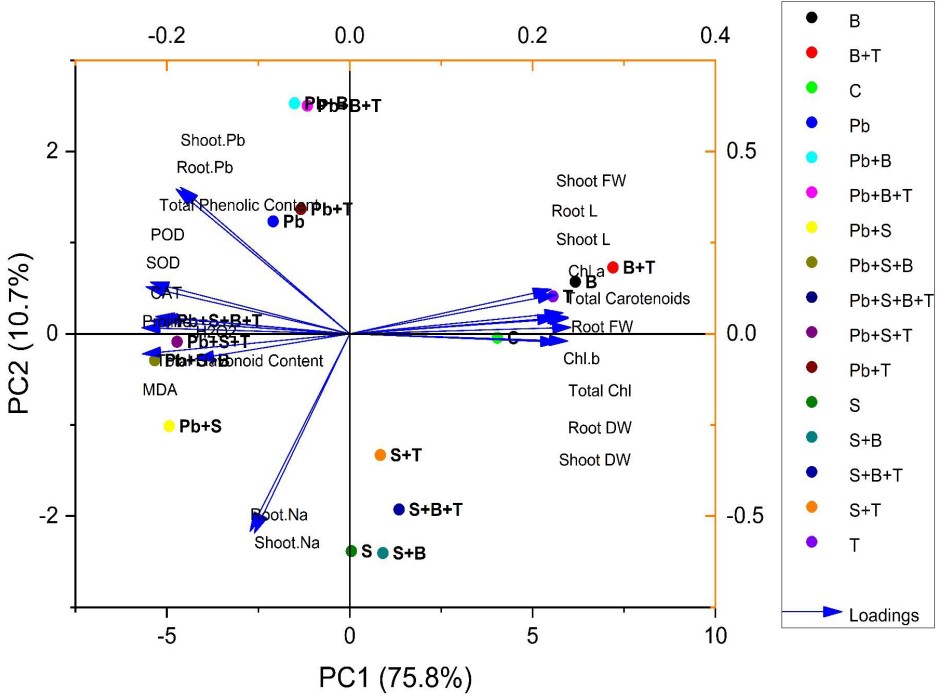

**Figure 3 Principal component analysis of different attributes of *H. annuus* plant.** Shoot length, Shoot L; Root length, Root L; Shoot fresh weight, Shoot FW; Root fresh weight, Root FW; Shoot dry weight, Shoot DW; Root dry weight, Root DW; Concentration of Lead in shoot, Shoot. Pb; Concentration of Lead in root, Root. Pb; Malondialdehyde, MDA; Hydrogen peroxide, $H_2O_2$; Peroxidase, POD; Superoxide dismutase, SOD; Catalase, CAT; Concentration of Sodium in shoot, Shoot. Na; Concentration of Sodium in root, Root. Na; Chlorophyll a, Chl. *a*; Chlorophyll b, Chl. *b*; and Total Chlorophyll, Total Chl.; C, control; B, bacteria; T, thiamine.

and salt) stress had a prominent beneficial effect on various growth characteristics of sunflower than the control. The variables aligned with PC1 were correlated positively, such as root L, shoot L, root FW, root DW, shoot FW, shoot DW, total carotenoids, total Chl., Chl. *a* and Chl. *b*. However, a highly negative relationship between the variables of PC1 and PC2 was observed: shoot Pb; root Pb; POD; SOD; CAT; total flavonoid content; total phenolic content; proline; MDA; $H_2O_2$; shoot Na and root Na (Fig. 3).

Treatments under no stress exhibit scores clustered within a relatively small range, having values such as C (4.03, −0.04), B (6.16, 0.57), T (5.55, 0.41), and B+T (7.19, 0.72). However, treatments under salinity stress (S) exhibit scores clustered within a small range with positive PC1 scores and negative PC2 scores, such as S (0.04, −2.38), S+B (0.90, −2.40), S+T (0.83, −1.33) and S+B+T (1.34, −1.92). Treatment under lead (Pb) stress showed negative PC1 scores and positive PC2 scores, such as Pb (−2.09, 1.23), Pb+B (−1.50, 2.52), Pb+T (−1.33, 1.36) and Pb+B+T (−1.15, 2.50). Meanwhile, treatment with combined Pb and salinity stress (Pb+S) showed negative scores for PC1 and PC2. Pb+S (−4.93, −1.01), Pb+S+B (−5.33, −0.29), Pb+S+T (−4.72, −0.08) and Pb+S+B+T (−5.004, 0.14). All the treatments showed great variability in the PCA biplot.

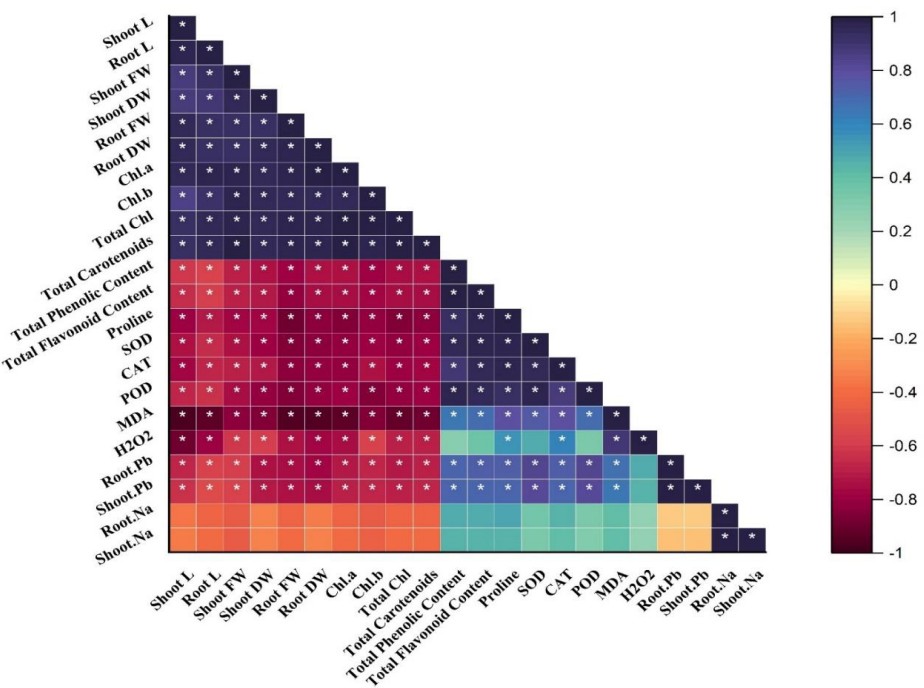

**Figure 4 Correlation matrix representing correlation among different attributes of H. annuus plant.** Shoot length, Shoot L; Root length, Root L; Shoot fresh weight, Shoot FW; Root fresh weight, Root FW; Shoot dry weight, Shoot DW; Root dry weight, Root DW; Concentration of Lead in shoot, Shoot. Pb; Concentration of Lead in root, Root.Pb; Malondialdehyde, MDA; Hydrogen peroxide, $H_2O_2$; Peroxidase, POD; Superoxide dismutase, SOD; Catalase, CAT; Concentration of Sodium in shoot, Shoot. Na; Concentration of Sodium in root, Root. Na; Chlorophyll a, Chl. *a*; Chlorophyll b, Chl. *b*; and Total Chlorophyll, Total Chl.; C, control; B, bacteria; T, thiamine. * depicts significance ($p \leq 0.05$) level.

## Correlation analysis

A Pearson correlation matrix was performed among the studied parameters of sunflower (Fig. 4) indicated by the color scale. Blue, showing a highly positive correlation, light yellow indicating no correlation between the parameters, and dark red showing a highly negative correlation between the parameters. Shoot length (shoot L) was negatively correlated with total phenolics content; total flavonoid content; proline content; SOD; CAT; POD; MDA; $H_2O_2$; root Pb; shoot Pb; shoot Na, and root Na. There was a slightly negative correlation between root Na, shoot Na, and root Pb, shoot Pb. On the other hand, shoot length (shoot L) was positively correlated with root L, shoot FW, shoot DW, root FW, root DW, and Chl. *a*; Chl. *b*; total Chl. and total carotenoids. All of these were significantly and positively correlated with each other as well. The treatments demonstrate (*) within rows that depict significance ($p \leq 0.05$) level.

## DISCUSSION

The increasing menace of salinity and heavy metal pollution in soil by natural and anthropogenic sources leads to disturbance in the crop production system by disrupting

soil quality. Many physical, chemical and biological remediation techniques are used to eliminate salt and Pb contamination from soil. One of the leading soil remediation techniques is Phyto-remediation. *B. subtilis* IAGS174 was checked for its potential to tolerate Pb and salt stress in this experiment. In our study, *B. subtilis* IAGS174 survived under different levels of NaCl. In addition, the isolated bacteria showed significant tolerance to Pb stress. In agreement with our results, *Daraz et al. (2023)* isolated salt and Cd-tolerant bacteria and observed that different bacteria might tolerate salt stress to varying levels, like *H. bacillus,* which tolerates up to 29% of NaCl.

Salinity and heavy metal contamination are the two most important threats to our environment, which are caused by natural and anthropogenic factors and pollute soil, water, and air. These pollutants affect soil fertility and can enter the food chain through crops. For the sustainable production of crops, there is a need to adopt more environmentally friendly remediation techniques that can eliminate these pollutants or at least reduce their uptake in edible parts. During the current study, Pb/salt-tolerant *B. subtilis* IAGS174 and thiamine were selected to reduce salinity and Pb toxicity in sunflower. The environmental variability and confounding factors may interfere with the study findings; hence, these were controlled by adapting a completely randomized research design, replication and precision during experimental setup, such as treatment application, as well as quality analysis to get valid and reliable results. Moreover, quality control was also exercised during chemical acquisition, sample preparation, plant analysis and data handling.

During the present study, the negative effects of Pb and salt on growth attributes and photosynthetic pigments of sunflowers relate to Pb and salt phototoxicity, ROS production, and disturbance in the metabolic functions of the concerned plants. Disturbance in chlorophyll production and the electron transport chain, with stomatal closure, also reduces plant growth and yield under stress conditions. The deceleration of the Calvin cycle, impeding disorder in mitochondrial activity and plastoquinone production, and disruption in $CO_2$ assimilation as well as stomatal conductance during toxic regimes could all be contributing factors to the decrease in plant growth and biomass (*Altaf et al., 2021*). *B. subtilis* IAGS174 secrete a variety of growth-promoting biochemicals that increase assisted plant growth and abiotic stress resistance in plants growing under stressed conditions. Previous studies proved that *B. subtilis* helps in the phytoextraction of heavy metals as it increases the stress tolerance of plants, which induces metal translocations in plant roots and shoots (*Ayub et al., 2024*). The bacteria perhaps improved the growth of the plant exposed to abiotic stress by producing 1-carboxylate (ACC) deaminase, auxin, siderophores, and Fe-proteins. The results of this experiment are in agreement with *Ayub et al. (2024)*, who stated a similar increasing trend for growth increment in Brassica by using a combined treatment of two *Pseudomonas* species (*Pseudomonas fluorescens* and *P. gessardi*) with compost under Cd and salt. Moreover, salt-tolerant Bacillus strains (NMCN1 and LLCG23) in wheat promoted the growth and photosynthetic activity under saline conditions (*Ayaz et al., 2022*). Additionally, Pb-tolerant *Bacillus* strains (*B. megaterium* N29 & *B. safensis* N11) improved the growth, photosynthetic, and antioxidant capability of spinach plants grown in sewage water (*Najm-ul-Seher et al., 2021*). PGPRs' growth-promoting attributes enable them to enhance the growth and phytoremediation of assisted plants (*Rao et al.,*

2025). Thiamine acts as a secondary metabolite and phytostabilizer because it regulates heavy metal translocation in plant roots and shoots. *Ahmed & Sattar (2024)* reported that thiamine dose enhanced the growth, photosynthetic pigments, and antioxidants in *Vicia faba* plants under saline conditions. *Alves et al. (2025)* observed that thiamine improved gas exchange attributes, growth, and biomass production in soybean plants. Thiamine used alone or in combination with alpha-tochopherol synergistically mitigated salt stress and improved root and shoot growth of sorghum (*Mohamed-Hussein & Orabi, 2024*). Thiamine may have improved salinity and Pb toxicity resilience in applied plants by promoting enzymatic and non-enzymatic activities, expressing stress-alleviating genes and accelerating photosynthetic machinery.

The results of our study depicted a noticeable decline in Chl. *a*, Chl. *b*, total chlorophyll and carotenoid contents in sunflowers under Pb and salinity stress. Heavy metals impede chlorophyll production by deactivating the enzymes involved in the biosynthesis of photosynthetic content. Changes in pigment content under stressful conditions directly affect plant resilience and photosynthate manufacture (*Trentin et al., 2025*). Secondly, Pb stress inhibits the inclusion of Fe in the Phyto-porphyrin ring of chlorophyll, which reduces chlorophyll synthesis and ultimately causes damage to chlorophyll molecules (*Bender et al., 2025*). Under salt stress, plants show a similar trend of decreasing pigment as the photosynthetic rate and stomatal conductance decrease continuously. So, under the combined stress of Pb and salt conditions, our experiment showed the lowest contents of chlorophyll pigments in *H. annuus*. A similar depression of chlorophyll pigments in *Spinacia oleracea* grown under Cd toxicity was observed by *Tanveer et al. (2022)*. *Siddika et al. (2024)* described that *B. subtilis* augmented the growth and photosynthetic pigments in rice plants grown in saline circumstances. During the present research, plants treated with thiamine and *B. subtilis* showed an increase in the biosynthesis of photosynthetic pigments because PGPR and thiamine can trigger plant defense systems against salt and Pb stress. Similarly, *Espinosa-Palomeque et al. (2025)* revealed that rhizobacteria enable plants to manage pathogens and improve stress resilience. Thiamine priming can activate the plant defense mechanism under abiotic stress. Thiamine is also known to scavenge or fight against ROS species and thus helps to sustain chlorophyll pigment levels in sunflower plants. Increased plant growth may have resulted from improved photosynthetic pigment production following rhizobacterial inoculation, which restored the plant's photosynthetic potential under Pb and salt stress. *Jabeen et al. (2022)* observed that thiamine application improved the growth, photosynthetic, and antioxidant parameters of *Brassica oleracea* exposed to arid conditions. *Yusof (2019)* found that exogenously applied thiamine improves plant resilience and nutritional quality. The rhizobacteria improved the Fe uptake, which helped in the chlorophyll synthesis by increasing the leaf area and ultimately enhanced the photosynthetic activity of plants under a stressed environment (*Khan et al., 2018*).

A higher amount of ROS generated under toxic regimes induces oxidative stress, damaging cellular membranes and DNA in plants (*Verma & Dubey, 2003*). The results of our study depicted a substantial rise in ROS-initiated oxidative stress in *H. annuus* under Pb and salinity toxicity. The content of antioxidant enzymes, *i.e.,* POD, SOD, and CAT, was found to be higher in Pb and salt-stressed sunflower as compared to controlled conditions

(Fig. 1). *Ahluwalia, Singh & Bhatia (2021)* observed that heavy metal stress increases the activity of different anti-oxidant enzymes in plants for capturing ROS. Moreover, *B. subtilis* and thiamine also showed higher antioxidant enzyme production in stressed sunflowers. High amounts of antioxidants are produced to counter the increased content of ROS in stressed plants. Our results are in agreement with *Gul-Lalay et al. (2024)*, who found that PGPRs and biochar boosted anti-oxidative enzymes in stressed plants. The increased activity of anti-oxidant enzymes by bacteria and thiamine decreased the negative effects of salt and Pb on sunflower by scavenging ROS. Augmented rate of SOD in plants induced tolerance against Pb and salt stress. It converts toxic $O_2^-$ radicals to oxygen molecules. SOD also regulated the intercellular ROS and other physiological conditions in plants exposed to abiotic stress (*Faize et al., 2011*). *Mohamed-Hussein & Orabi (2024)* exhibited that thiamine supply diluted the salinity-induced oxidative impairments in sorghum plants by increasing the antioxidant enzymes such as SOD, CAT, APX, GR, and POD. Moreover, thiamine dose improved the enzymatic (SOD, CAT, POD) and non-enzymatic antioxidants (phenol, flavonoid, proline) in maize plants grown in arsenic (As) stress (*Atif et al., 2022*). Analogous to our investigations, *Jamil et al. (2024)* observed that inoculation of *B. cereus* and *B. aerius* augmented the antioxidant enzymes such as SOD, POD, and CAT in spinach plants grown under heavy metal stress.

During our study, phenol, flavonoid, and proline showed an increasing trend in *H. annuus* under Pb and salt stress (Table 2). Higher levels of osmoprotectants including soluble sugars, proteins, phenols, flavonoids, proline, and amino acids, accumulate in plants exposed to abiotic stress (*Kanwal et al., 2024*). The increased rate of flavonoids, phenols, and proline demonstrates the ability of *H. annuus* to withstand Pb and salt stresses (*El-Tayeh, Youssef & Hassan, 2023*). Due to this, proline is considered an important osmoregulator for many plants to eliminate salt and Pb stress. Proline serves as the primary source of energy and nitrogen and stabilizes the macromolecular structure. In a similar study, an increased amount of proline was depicted in the canola plant under salt stress compared to non-stressed conditions (*Vazayefi et al., 2024*).

Using PGPRs and thiamine, sunflower plants may enhance stress tolerance due to up-regulation in soluble protein, proline, phenol, flavonoids, and antioxidant enzyme activity. Proline, phenol, flavonoids, and soluble carbohydrates contribute to membrane integrity and endow stress resistance in plants. By preventing tissue damage, these systems enable the plant to grow and develop in saltwater and heavy-metal-polluted environments. Our results are in agreement with *Atif & Perveen (2024)*, who depicted that the combined effect of IAA and thiamine increased the amount of proline in stressed plants. Similarly, our study showed higher proline contents in plants under the combined effect of *B. subtilis* and thiamine. Exogenously applied thiamine augmented stress tolerance in *Pisum sativum* by increasing growth, photosynthesis, antioxidant enzymes, phenolics, and proline (*Kausar et al., 2023*). By our findings, *Khan, Umar & Iqbal (2023)* revealed that supplementation of *P. fluorescens* (NAIMCC-B-00340) and *Azotobacter chroococcum* Beijerinck 1901 (MCC 2351) alleviated salinity stress in *Pusa Jagannath* plants by increasing antioxidant enzymes, phenol, flavonoid, and proline contents.

In our study, sunflower plants exhibited higher contents of both $H_2O_2$ and MDA when exposed to Pb and salinity toxicity (Fig. 2). The overabundance of these two biomolecules interfered with the functioning of the cell membrane (*Bhat et al., 2022*). In contrast, MDA and $H_2O_2$ levels significantly declined in *B. subtilis* and thiamine-supplied sunflowers under Pb and salt stress. Our results are in agreement with *Ayub et al. (2024)*, who observed the reduced production of MDA, electrolyte leakage, and $H_2O_2$ in PGPR and compost-supplied plants under stress. The limited production of MDA and $H_2O_2$ isalso associated with the reduced uptake of Pb and Na in thiamine-applied *H. annuus*. Plants treated with thiamine activated their defense mechanisms and restricted the amounts of both MDA and $H_2O_2$, thereby reducing the harmful stress effects. Analogous to our findings, *Kaya et al. (2020)* found that thiamine application significantly ameliorated boron-induced oxidative stress by decreasing $H_2O_2$, electrolyte leakage and MDA levels in pepper plants. Similarly, exogenous application of thiamine improved physiochemical activities, leading to drought stress mitigation in Xinjiang cotton (*Zhao et al., 2025*). Moreover, in line with our results, *Sehrish et al. (2024)* found that PGPR-assisted *Triticum aestivum* L. plants exhibited lower levels of MDA, electrolyte leakage, and $H_2O_2$ when raised under Cd stress.

During our study, it was noted that the amount of Pb and Na concentration was increased in roots and shoots in *H. annuus* when treated with *B. subtilis* IAGS174. Our results are in agreement with those of *Bender et al. (2025)*, who noticed the augmented accretion of the metal in microbe-assisted sunflowers under Pb stress. Pb-tolerant rhizobacteria might decrease soil pH, which is important for the solubility and bioavailability of metal. Through alterations in Pb and Na availability and solubility, in addition to redox fluctuations in the rhizosphere, *B. subtilis* IAGS174 assisted sunflower to exhibit higher Na and Pb uptake in its root and shoot tissues. In addition to acidifying the soil, plant roots release protons and organic acids that reduce heavy metal adsorption and promote metal mobility in the rhizosphere. On the other hand, thiamine supply reduced the uptake of Na and Pb in the roots and shoots of sunflower plants (Table 3). It may be assumed that thiamine uptakes and immobilizes the Na and Pb in roots and restricts the translocation to other plant parts.

In our study, the rate of translocation and bioconcentration factor varied by *B. subtilis* IAGS174 and thiamine application to *H. annuus* under Pb and salt stress. Na and Pb-translocation increased in plants from roots to shoots when supplemented with *B. subtilis* IAGS174 (Table 3). The translocation of heavy metals in plants depends upon soil chemistry. So, the application of *B. subtilis* IAGS174 accelerated the extractable amount of Pb and salt in the soil, which led to augmented uptake in roots and shoot tissues of sunflower plants. Analogous to our findings, it was reported that Plant growth-promoting *Micrococcus luteus* WN01 alleviated stress and improved phytoextraction capability of *Ocimum gratissimum* growing under combined stress of Cd and crude oil (*Choden et al., 2025*). A higher amount of Pb and salinity was found in the shoots than in the roots. Nevertheless, *B. subtilis* IAGS174-assisted plants exhibited an increased rate of bioconcentration and translocation of Na and Pb. The metal accumulation was more in shoots than roots, corroborating the study of *Atif et al. (2022)*, which showed more translocation of Arsenic in shoots. Microbial association improves phytoextraction capability of oilseed crops (*Kowalska & Biczak, 2025*). Conversely, a decrease in bioconcentration and translocation rate was observed in

sunflower plants treated with thiamine (Table 3). Thiamine as a phytoregulator improved the tolerance capacities of plant roots and immobilized Na and Pb in roots, making it impossible to translocate to other parts of the plant. Similarly, thiamine application decreased the Pb and Na accumulation in roots and shoots of *Lens culinaris* exposed to Pb stress (*Bouhadi et al., 2024*). Moreover, *Ahmed & Sattar (2024)* reported that thiamine supply lowered the Na uptake in *V. faba* plants treated with salinity toxicity. Furthermore, thiamine supplementation reduced the uptake and translocation of Cd in the root and shoot of canola plants under Cd stress (*Sanjari et al., 2019*). The combined treatment of thiamine and bacteria showed an improved rate of translocation and bioconcentration in *H. annuus,* as the plant is also a hyperaccumulator and has a higher capability for metal uptake (*Naheed et al., 2022*).

## CONCLUSION

Results of the current study revealed that the salt and Pb stress significantly decreased plant growth, biomass production and photosynthetic pigments. Moreover, salt and Pb stress significantly accelerated the levels of antioxidant enzymes, proline, flavonoids, phenolic contents, MDA and $H_2O_2$. On the other hand, supplementation of *B. subtilis* IAGS174 and thiamine synergistically enhanced the growth, biomass and photosynthetic activity of *H. annuus* under single as well as combined salt and Pb stress and endowed stress resilience in sunflower. Anti-oxidant enzymes, proline, flavonoid, and phenolic content also increased in bacteria and thiamine-treated *H. annuus* exposed to Pb and salt stress. Application of *B. subtilis* IAGS174 and thiamine successfully reduced MDA and $H_2O_2$ contents in *H. annuus* plants under Pb and salinity stress, which might have alleviated oxidative stress in plants. Our results exhibit that *B. subtilis* IAGS174 enhanced phytoextraction of Na and Pb. Thiamine, on the other hand, acted as a phytostabilizing agent and decreased the uptake of Na and Pb. Hence, these two agents offered a balanced stratagem for phytoremediation and stress mitigation in a hyperaccumulator plant. The synergistic supply of *B. subtilis* IAGS174 and thiamine may be a suitable, cheap and eco-friendly alternative for ornamental crops growing in multi-polluted soil. In the future, proteomics, genomics and metabolomics studies may assist in understanding plant stress response. Furthermore, field trials are mandatory to evaluate the efficacy of the microbe-thiamine combination under natural stress conditions for sustainable and eco-friendly crop production.

### Funding
The authors received no funding for this work.

### Competing Interests
Nasim Ahmad Yasin is an Academic Editor for PeerJ.

## Author Contributions

- Sidra Fatima conceived and designed the experiments, performed the experiments, analyzed the data, prepared figures and/or tables, authored or reviewed drafts of the article, and approved the final draft.
- Waheed Ullah Khan conceived and designed the experiments, performed the experiments, analyzed the data, authored or reviewed drafts of the article, and approved the final draft.
- Rehana Sardar conceived and designed the experiments, performed the experiments, analyzed the data, prepared figures and/or tables, and approved the final draft.
- Bareera Munir conceived and designed the experiments, performed the experiments, analyzed the data, prepared figures and/or tables, authored or reviewed drafts of the article, and approved the final draft.
- Areeba Rehman conceived and designed the experiments, performed the experiments, analyzed the data, prepared figures and/or tables, authored or reviewed drafts of the article, and approved the final draft.
- Waheed Akram conceived and designed the experiments, performed the experiments, analyzed the data, prepared figures and/or tables, authored or reviewed drafts of the article, and approved the final draft.
- Iqra Munir conceived and designed the experiments, performed the experiments, analyzed the data, prepared figures and/or tables, authored or reviewed drafts of the article, and approved the final draft.
- Nasim Ahmad Yasin conceived and designed the experiments, analyzed the data, prepared figures and/or tables, authored or reviewed drafts of the article, and approved the final draft.

## Data Availability

    The raw measurements are available in the Supplemental File.

## Supplemental Information

Supplemental information for this article can be found online at http://dx.doi.org/10.7717/peerj.19527#supplemental-information.

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
