# Peer review of "Synergistic application of Bacillus subtilis IAGS174 and thiamine to mitigate salinity and lead stress in Helianthus annuus"

_PeerJ, doi:10.7717/peerj.19527_

## Round 0.1 · original submission · Major Revisions

Abiotic stress caused by salinity and lead is an important phenomenon we should find how oil crops like sunflowers respond to this adverse environmental condition and how we reduce the negative impact of it. Your research involves valuable insights to mitigate its unfavorable impacts. Nevertheless, it is essential to address certain technical details to enhance the article further. I encourage you to carefully review the reviewers' suggestions and thoughtfully consider each recommendation. If you find yourself in disagreement with any specific suggestions, providing a clear and well-supported rationale for your viewpoint would be highly beneficial.

**Language Note:** The review process has identified that the English language must be improved. PeerJ can provide language editing services - please contact us at [email protected] for pricing (be sure to provide your manuscript number and title). Alternatively, you should make your own arrangements to improve the language quality and provide details in your response letter. – PeerJ Staff

·

Basic reporting

See attached report

Experimental design

See attached report

Validity of the findings

See attached report

Additional comments

See attached report

·

Basic reporting

• The manuscript contains numerous grammatical errors, awkward phrasing, and unclear sentences. The readability is significantly hindered, making it difficult to follow the argument. A professional English language revision is necessary before further consideration.
• While the introduction provides some background, the knowledge gap is not clearly defined. The authors should explicitly highlight how this study differs from existing literature. Additionally, the literature review appears biased toward certain sources, with limited discussion of conflicting or alternative viewpoints.
• Some figures are poorly labeled and lack clear descriptions. The quality of images and graphs is subpar, making it difficult to interpret trends. Statistical details (e.g., error bars, p-values) are either missing or inadequately explained.

Experimental design

• The study lacks a well-articulated hypothesis. The authors must clearly define their research question and its novelty.
• The experimental design lacks proper controls. Without adequate controls, the validity of comparisons between treatments is questionable. The sample size is not justified, raising concerns about statistical power. how was the bacterial strain identified and characterized? This omission weakens the study's credibility. how was the foliar spray concentration determined? Was there a dose-response study?

Validity of the findings

• The statistical analysis is poorly described, and it is unclear if the appropriate tests were used. The authors should provide clear justifications for statistical methods and include proper error analysis.
• Several claims are not supported by sufficient data. For example, the claim that "Bacillus subtilis IAGS174 accelerated translocation of Pb and Na" lacks a mechanistic explanation or supporting evidence. Correlation does not imply causation—several conclusions are overstated without ruling out alternative explanations.
• The authors do not address potential confounding factors, such as environmental variability. There is no discussion on the applicability of results to field conditions, making it unclear if the findings are relevant beyond controlled environments.

Additional comments

• The overall structure is weak, with abrupt transitions between sections. The discussion should be more logically structured, ensuring that each point builds upon the previous one.
• The conclusions are overstated and not fully supported by data. The authors should tone down strong claims and acknowledge uncertainties.
• The manuscript does not provide practical implications for agriculture. The authors should clarify how their findings can be applied in real-world settings.

Reviewer 3 ·

Basic reporting

Review
Title: Synergistic application of Bacillus subtilis IAGS174 and thiamine to mitigate salinity and
lead stress in Helianthus annuus
Manuscript Number. 113498

Review Report
Title: The title is appropriate.
Abstract: The abstract is well written. However, it should include detailed parameters, specifying their values increase or decrease in % with the applied treatments. For instance, include details on parameters such as shoot length, root length, fresh and dry weights of root and shoot, chlorophyll a, chlorophyll b content, etc.
Introduction. The introduction is well written but slightly lengthy. It should be condensed to fit within two pages.
Materials and Methods:
1. Specify the name, model, and specifications of the equipment used to estimate photosynthetic pigments.
2. Provide the name, model, and specifications of the equipment used to estimate flavonoid and phenolic content.
3. Mention the name, model, and specifications of the equipment used to estimate proline content
4. In line 203, specify the model and specifications of the flame photometer.
Statistical Analysis: The statistical analysis is well described.
Results: The results are well written.
Discussion: Reduce the length of the discussion, as it is too lengthy.
Conclusion: The conclusion is too generic and should be more specific.

Experimental design

Experimental is well described.

Validity of the findings

This study is valid as it investigates the synergistic role of Bacillus subtilis IAGS174 and thiamine in mitigating salinity and lead stress in Helianthus annuus, addressing a critical issue in plant stress physiology. The findings demonstrate enhanced growth, antioxidative defense, and stress tolerance mechanisms, supporting sustainable agricultural practices. This research provides a scientifically sound and practical approach for improving crop resilience in contaminated soils.

Additional comments

This study presents a well-structured and scientifically significant approach to addressing abiotic stress in Helianthus annuus using Bacillus subtilis IAGS174 and thiamine. The experimental design, including controlled stress conditions and biochemical assessments, strengthens the reliability of the findings. The research not only highlights the role of plant growth-promoting rhizobacteria (PGPR) and thiamine in stress mitigation but also offers practical implications for sustainable agriculture. Future studies could explore the long-term field applicability of this approach and its effects on other crops to further validate its agricultural significance.

---

## Round 0.2 · Minor Revisions

Abiotic stress caused by salinity and lead is an important phenomenon we should find how oil crops like sunflowers respond to this adverse environmental condition and how we reduce the negative impact of it. Your research involves valuable insights to mitigate its unfavorable impacts. Nevertheless, it is essential to address certain technical details to enhance the article further. I encourage you to carefully review the reviewers' suggestions and thoughtfully consider each recommendation. If you find yourself in disagreement with any specific suggestions, providing a clear and well-supported rationale for your viewpoint would be highly beneficial.

·

Basic reporting

Satisfactory

Experimental design

Satisfactory

Validity of the findings

Satisfactory

Additional comments

NC

·

Basic reporting

The manuscript suffers from several critical shortcomings that compromise its scientific rigor, clarity, and contribution to the field.
The following key issues highlight the major flaws in the study:
• The research question is poorly justified, and the study does not offer a novel contribution to the existing body of knowledge.
• The literature review is insufficient, lacking a comprehensive discussion of recent and relevant studies.
• The study appears to replicate previous findings without introducing significant innovation or new insights.
• The methodology is inadequately described, making replication difficult.
• The sample size is too small to draw meaningful conclusions, and statistical validation is weak.
• The experimental design lacks proper controls, leading to questionable reliability.
• Key parameters and variables are either not defined or inconsistently reported.
• The data analysis is superficial and does not provide a strong basis for the conclusions drawn.
• Statistical tests are either inappropriate or poorly applied, leading to misleading interpretations.
• The results are not presented logically, making it difficult to follow the study’s findings.
• The discussion section lacks depth and does not critically engage with the results in the context of existing literature.
• Figures and tables are not well-integrated into the discussion, and some data appear redundant or irrelevant.
• The authors overstate the significance of their findings without sufficient supporting evidence.
• The manuscript contains numerous grammatical and typographical errors, which hinder readability.
• The writing lacks coherence, and many sections are either repetitive or unclear.
• Formatting inconsistencies in citations, references, and section headings make the manuscript difficult to follow.
• The conclusion is vague and does not adequately summarize the key findings.
• The study's limitations are not acknowledged, giving a misleading impression of the research's reliability.
• No clear recommendations for future research are provided.
Due to the significant methodological flaws, lack of originality, and weak data analysis, this manuscript is not suitable for publication in its current form. Major revisions are necessary to improve the clarity, rigor, and scientific contribution of the study.
Recommendation: Major Revisions Required

Experimental design

See details in basic reporting

Validity of the findings

See details in basic reporting

Additional comments

See details in basic reporting

·

Basic reporting

Review the spelling of the text.

Experimental design

Factivel with the study.

Validity of the findings

Factivel with the study.

Additional comments

The article entitled "Synergistic application of Bacillus subtilis IAGS174 and thiamine to mitigate salinity and lead stress in Helianthus annuus" has been revised. Some suggestions are highlighted and described in the text. After some changes this manuscript is a candidate for publication.

---

## Round 0.3 · accepted · Accept

I would like to thank you for accepting the referees' suggestions and improving your article based on their suggestions. Your article is ready to publish. We look forward to your next article.

·

Basic reporting

The manuscript lacks a clear and professional structure. Essential sections such as the introduction, methods, results, and discussion are not distinctly separated, making it difficult to follow the research flow.

There is insufficient background provided. The current literature review does not adequately frame the research problem or highlight the knowledge gap the study intends to address.

Figures and tables are poorly integrated and lack proper labeling. Additionally, there is no mention of raw data availability, which is necessary for reproducibility and transparency.

Several sections contain ambiguous phrasing and grammatical errors that impede comprehension. A thorough language revision is required.

Experimental design

The hypothesis and research objectives are not clearly defined. It remains unclear how the study fills a specific knowledge gap or contributes to the existing body of literature.

Methods are inadequately described. Key experimental procedures lack sufficient detail to allow replication, and there is no mention of ethical approvals where applicable.

The study appears to deviate from the journal’s aims and scope, as the experimental work does not align clearly with the journal’s focus areas.

Validity of the findings

There is a lack of supporting data for key conclusions. Statistical analyses are either absent or insufficiently described, raising concerns about the reliability of the results.

The conclusions drawn are not adequately supported by the presented data. Several claims seem speculative without clear evidence linking them to the experimental outcomes.

Additional comments

The current manuscript has several fundamental issues in reporting, methodology, and data validity. Without substantial revisions addressing these concerns, the manuscript does not meet the journal’s standards for publication.

·

Basic reporting

Satisfactory.

Experimental design

Satisfactory.

Validity of the findings

Satisfactory.

Additional comments

See details in basic reporting.